# Experimental Study on Diesel Engine Emission Characteristics Based on Different Exhaust Pipe Coating Schemes

**DOI:** 10.3390/mi12101155

**Published:** 2021-09-25

**Authors:** Keqin Zhao, Diming Lou, Yunhua Zhang, Liang Fang, Yuanzhi Tang

**Affiliations:** School of Automobiles Studies, Tongji University, Shanghai 201804, China; zhaokeqin@tongji.edu.cn (K.Z.); loudiming@tongji.edu.cn (D.L.); fangliang@tongji.edu.cn (L.F.); tangyuanzhi@tongji.edu.cn (Y.T.)

**Keywords:** diesel engine bench test, basalt fiber, glass fiber, thermal insulation performance, emission characteristics

## Abstract

The thermal insulation performance of exhaust pipes coated with various materials (basalt and glass fiber materials) under different braiding forms (sleeve, winding and felt types) and the effects on the emission characteristics of diesel engines were experimentally studied through engine bench tests. The results indicated that the thermal insulation performance of basalt fiber was higher than that of glass fiber, and more notably advantageous at the early stage of the diesel engine idle cold phase. The average temperature drop during the first 600 s of the basalt felt (BF) pipe was 2.6 °C smaller than that of the glass fiber felt (GF) pipe. Comparing the different braiding forms, the temperature decrease in the felt-type braided material was 2.6 °C and 2.9 °C smaller than that in the sleeve- and winding-type braided materials, respectively. The basalt material was better than the glass fiber material regarding the gaseous pollutant emission reduction performance, especially in the idling cold phase of diesel engines. The NO_x_ conversion rate of the BF pipe was 7.4% higher than that of the GF pipe, and the hydrocarbon (HC) conversion rate was 2.3% higher than that of the GF pipe, while the CO conversion rate during the first 100 s was 24.5% higher than that of the GF pipe. However, the particulate matter emissions were not notably different.

## 1. Introduction

Diesel engines are widely applied in the field of commercial vehicles due to their good power features, fuel economy, and reliability. Although they have many advantages, they have a significant impact on the problem of environmental pollution worldwide [1,2]. The nitrogen oxide (NO_x_) and particulate matter emissions of diesel engines are considerable, accounting for 70% and 90%, respectively, of the total vehicle emissions, and are also responsible for several health problems [3,4]. To limit the pollutant emissions originating from diesel vehicles, various countries have continuously tightened emission limits, broadened the requirements for low-temperature and low-load emission pollutant control, and greatly reduced the NO_x_ emission concentration limit. The combination of various after-treatment technologies has become an important means to improve vehicle pollutant emissions and meet the increasingly stringent emission regulations. At present, after-treatment devices largely include optimized combustion + selective catalytic reduction (SCR), and exhaust gas recirculation (EGR) + diesel oxidation catalyst (DOC) + diesel particulate filter (DPF) [5,6,7,8]. The exhaust temperature is the main factor influencing the purification effect of the exhaust gas thermal reaction and the after-treatment performance. Within a certain temperature range, the higher the exhaust temperature, the better the purification effect of the thermal reaction [9]. If the exhaust temperature is too low, this may cause problems such as urea solution crystallization in the SCR device [10,11]. Therefore, to improve the catalytic efficiency of after-treatment devices, it is necessary to coat exhaust pipes with an insulating material to reduce heat loss, thereby improving the emission performance of the entire vehicle.

At present, glass fiber and asbestos fiber are mainly used as the insulation materials in the exhaust pipe, but long-term exposure to glass fiber and asbestos fiber will damage the respiratory system and cause health hazards [12,13]. Therefore, it is very important to study new alternative materials. Basalt fiber has increasingly become a substitute for glass fiber in many fields, such as marine and military industries, due to its excellent performance. Additionally, BF is labeled as safe, according to both the USA and the European occupational safety guidelines.

Basalt fiber is a new natural green material. It consists of volcanic extruded rock. Its chemical composition is similar to that of gabbro. The SiO_2_ content varies between 45% and 60%. The K_2_O + Na_2_O content is slightly higher than that in intrusive rocks. The Fe_2_O_3_ + FeO and MgO contents are slightly lower than those in intrusive rocks. Fiber is produced by putting material into a furnace where it is melted at 1450–1500 °C. Subsequently, the molten material is forced through a platinum/rhodium crucible bushing to create fibers. Compared with glass fiber, basalt fiber is cheaper to produce due to less energy consumed and no additives being required. Additionally, it attains a better high-temperature resistance than glass fiber. Basalt fiber achieves a wider temperature range, from −200 to 800 °C, while the temperature range attained by glass fiber varies between approximately −60 °C and 450 °C [14]. At a working temperature of 400 °C, the breaking strength of basalt fiber can be maintained at 85%, while at a working temperature of 600 °C, its breaking strength can still be maintained at 80%. Moreover, if basalt fiber is pretreated at a temperature ranging from 780 to 820 °C, it can be applied at 860 °C without shrinkage [15]. Due to its excellent high-temperature resistance, basalt fiber has been widely adopted in transportation infrastructure, environmental protection and other fields [16,17,18,19,20,21].

In recent years, scholars have conducted research on the thermal properties of basalt and glass fiber materials. Their research results indicated that basalt fiber contained a large number of micropores to prevent air convection and heat radiation, and its thermal insulation performance was higher than that of glass fiber [22]; under exposure to the same radiant heat flux, basalt fiber material reached a higher temperature faster than glass fiber material due to its higher thermal emissivity [23]; mass loss occurred in the temperature range of 200 to 350 °C, the thermal decomposition temperature of basalt fiber was 40 °C higher than that of glass fiber, and basalt fiber reached a higher thermal stability [24]. Therefore, basalt fiber provides certain advantages over glass fiber in terms of its thermal performance, and as it is widely used in car mufflers and other parts in the automotive industry [25], basalt fiber exhibits a certain potential in diesel engine thermal management applications. However, there are still few studies on its application in the after-treatment thermal management of diesel engines.

This paper relies on diesel engine bench tests to assess the thermal insulation performance of different exhaust pipe covering schemes considering basalt and glass fiber materials. Moreover, various basalt fiber weaving forms (basalt sleeve (BS), basalt winding (BW), and basalt felt (BF)) and their influence on the emission characteristics of diesel engines are investigated herein.

## 2. Methodology

### 2.1. Test Materials

The test exhaust pipe samples adopted basalt sleeve (BS), basalt winding (BW), basalt felt (BF), and glass fiber felt (GF) with the equal thickness and bulk density. The variety of covering samples is presented in Table 1.

### 2.2. Test Device and Data Processing

The test device mainly included a diesel engine for the test and the bench test system. The system consisted of a Horiba 7200D and an AVL489. The Horiba 7200D was used to measure the CO, THC, NOx, etc., and the AVL489 was used to measure the particle emissions. Table 2 lists the technical parameters of the diesel engine type adopted in the test, and Table 3 summarizes the composition and model of the sampling equipment of the diesel engine bench test system.

Through the bench test, the original emission data of different pollutants before entering the after-treatment system, and the emission data of different pollutants through the after-treatment system with different covering schemes, was obtained. Finally, the conversion rate of different pollutants was determined, and the calculation method is shown in Equation (1) where PE1 is the original emission data, PE2 is the emission data after, and CR is the conversion rate of different pollutants.
(1)$$\mathrm{CR}=\frac{\mathrm{PE}1-\mathrm{PE}2}{\mathrm{PE}1}\times 100\%$$

### 2.3. Test Conditions and Plan

The cold test of the diesel engine bench adopted the World Harmonized Transient Cycle (WHTC) heavy-duty diesel engine test cycle. The WHTC cycle is a transient operating condition lasting 1800 s that changes at 1 s intervals. The cycle is divided into three stages, namely, 0~600 s as the cold operation stage, 600~1200 s as the transition stage, and 1200~1800 s as the hot operation stage, as shown in Figure 1.

The exhaust pipes were coated with basalt and glass fiber materials with equal thickness and bulk density. The glass fiber material adopted the felt-type weaving method. The basalt fiber material was subjected to three different weaving methods, i.e., sleeve, winding and felt types. The temperature at the outlet end of the diesel engine, and at the inlet of the after-treatment system, was measured under cycling conditions. The heat preservation characteristics of the entire exhaust system were studied, and the emission characteristics (NO_x_, CO, HCs, and PN) of the diesel engine containing an exhaust system coated with the above two materials under the same cycle, were analyzed. The test plan is listed in Table 4, and the layout of the test bench and temperature measurement points are shown in Figure 2.

## 3. Results and Discussion

### 3.1. Exhaust Temperature Analysis of the Full Cycle

Figure 3 shows the exhaust gas temperature T1 at the rear end of the vortex over time for the different exhaust pipe covering schemes of the tested diesel engine. In essence, the temperature measured at the T1 measurement point of the exhaust pipes with the different materials and covering methods was basically the same, because it was the outlet temperature of the turbocharger and was less affected by the insulation material. The average temperature at each stage during cold, transition and hot operation exhibited a stepwise upward trend, at 201 °C, 258 °C and 328 °C, respectively.

T2 is the temperature at the inlet of the after-treatment system, and its specific changes throughout the cycle are shown in Figure 4. At the T2 measurement point, the exhaust temperature of the exhaust pipes with the different materials and covering methods greatly varied (especially in the cold cycle). The exhaust temperatures of the BS and BW pipes at the cold operation stage were lower than those of the other pipes. Their average temperatures were 3 °C and 2 °C, respectively, lower than the overall average temperature in the cold phase.

This test considered T1-T2 (Δ T) to intuitively simulate the thermal insulation performance of the vehicle exhaust pipe system. Figure 5 shows the evolution of the temperature decrease in the exhaust pipes with different covering schemes. The overall exhaust pipe temperature experienced a downward trend. This mainly occurred because the exhaust pipe operated in the cold state upon cold start initiation. Some of the heat flux preheated the exhaust pipe, resulting in a notable heat energy loss, therefore the temperature decline was large. With increasing exhaust pipe temperature, the heat loss attributed to preheating gradually decreased, reducing the temperature drop. Figure 6 shows the average temperature drop considering the different covering schemes for the entire circulating exhaust pipe. The results indicated that the average temperature decrease in the BF pipe during the entire cycle was the smallest, which was 0.5 °C smaller than that in the GF pipe. This occurred because basalt fiber is composed of tectosilicates, phyllosilicates, chain silicates, and orthosilicates [26]. The amorphous region in the interior was large, and many grain boundaries, defects and impurities existed. This resulted in a low thermal conductivity. Therefore, the thermal insulation performance of the basalt fiber material was better than that of the glass fiber material. Comparing the different basalt material weaving methods, the average temperature decrease in the exhaust pipe coated with BF was 2.6 °C and 2.9 °C smaller than that in the exhaust pipes coated with BS and BW, respectively, and the heat preservation performance was higher than that of the other two weaving forms. This occurred due to the large size of the pores of the sleeve- and winding-type materials, resulting in serious heat loss and a poor thermal insulation performance.

### 3.2. Emission Analysis of the Full Cycle

Figure 7 shows a comparison of the NO_x_ conversion efficiency of the exhaust pipes between the different covering schemes during each cycle of the diesel engine. The figure reveals that with an increasing load, the NO_x_ conversion rate gradually increased. Figure 4 shows that the exhaust gas temperature rose, thus promoting SCR. During the first 600 s of the cold-state operation stage, the SCR inlet temperature was low, and the NO_x_ conversion rate of each pipe did not exceed 50% [27]. At the transition stage (600~1200 s) and hot-state operation stage (1200~1800 s), the NO_x_ conversion rate of each pipe was greatly improved; this occurred because DOC in the range of 200~400 °C can effectively improve the NO_2_/NO_x_ ratio and improve the NO_x_ conversion rate [28]. Regardless of the operation stage, the average NO_x_ conversion rate of the BF pipe was the highest. At the cold operation stage, the average NO_x_ conversion rate of the BF pipe was 47.6%, 97.6% and 7.4% higher than that of the BS, BW and GF pipes, respectively. At the transition stage (600~1200 s) and hot operation stage (1200~1800 s), the average NO_x_ conversion rate was 13.6%, 3% and 11% higher, respectively, than that of the other three pipes.

Figure 8 shows the average CO conversion rate of the exhaust pipes with the different covering schemes during the WHTC cycle. Under the emission reduction effect of the DOC and DPF device combination, the CO conversion rate of each pipe was maintained at a high level [29], reaching above 85%, and with the increase in exhaust temperature, CO oxidation was promoted and CO conversion rate increased [30]. The average CO conversion rate of the BF pipe was 1.1%, 2.7% and 4.3% higher than that of the BS, BW and GF pipes, respectively, but these differences were small.

Figure 9 shows the HC conversion efficiency of the exhaust pipes with the different covering schemes at each cycle stage. The figure revealed that the overall HC conversion efficiency gradually increased. The average HC conversion rate of the BF pipe reached 96.7%, which was 6.9%, 8.3% and 1.3% higher than that of the BS, BW and GF pipes, respectively.

Figure 10 shows the change curve of the particulate matter concentration of the exhaust pipes with the different covering schemes at each cycle stage. Under the effects of the DOC and DPF device combination, the particulate matter emissions of the different insulated pipes were relatively low and exhibited a trend of decreasing first and then slightly increasing. This occurred because at the beginning of the cycle, the exhaust temperature was low and the DOC performance was not high. During hot-state operation, a high exhaust temperature promoted the production of sulfate [31], thus causing a slight increase in particulate matter emissions.

Figure 11 shows the total emissions of gaseous pollutants and particulates during the WHTC cycle of the exhaust pipes with different covering schemes. It is observed that little difference existed in particulate matter emissions. Due to its better thermal insulation performance, the BF pipe achieved a high after-treatment system conversion efficiency, and the gaseous pollutant emissions were the lowest. The total emissions were 67.3%, 49.8% and 23.2% lower than those of the BS, BW and GF pipes, respectively.

### 3.3. Exhaust Temperature Analysis during Cold Operation

Since the WHTC cycle focuses more on the investigation of diesel engine emissions under low-speed and low-load conditions, the diesel engine emission temperature during this cycle is low, and the performance requirements of the SCR after-treatment system are high. Therefore, it is very important to study the temperature and corresponding emission characteristics of cold-state operation. Based on the test results, the cold operation stage (0–600 s) was subdivided, and the change law of the average temperature drop at each substage of the initial cold operation stage was further analyzed.

Figure 12 shows the average temperature drop of the exhaust pipes with the different covering schemes at the various substages of the initial cold-state operation stage. During the cold operation cycle of the diesel engine, the average temperature decrease in the BF pipe over the first 600 s was 2.6 °C smaller than that in the GF pipe.

### 3.4. Analysis of the Emission Results during Cold Operation

Figure 13 shows a comparison of the NO_x_ conversion efficiency of the exhaust pipes between the different covering schemes at each stage of cold operation. It is observed that the NO_x_ conversion rate during the first 100 s was lower than 20%. This finding matches the temperature characteristics. Figure 4 shows that during the first 100 s, the average exhaust gas temperature entering the after-treatment system was lower than 200 °C, resulting in a low SCR catalytic activity and a low NO_x_ conversion rate. The discharge performance of the BF pipe was the highest during cold-state operation, and the average NO_x_ conversion efficiency was 47.6%, 97.6%, and 7.4% higher than that of the BS, BW and GF pipes, respectively.

Figure 14 shows the CO conversion efficiency of the exhaust pipes with the different covering schemes at each stage during cold operation. Figure 14 shows that the CO conversion rate was low during the first 100 s and remained high during the next 500 s. The discharge effect of the BF pipe during the first 100 s was obviously better than that of the GF pipe, and the CO conversion rate of the BF pipe was 24.5% higher than that of the GF pipe.

Figure 15 shows a comparison of the HC conversion efficiency of the exhaust pipes between the different covering schemes at each stage during cold operation. The figure revealed that the HC conversion rate at each stage was high, being not lower than 75%. The HC conversion rate of the BF pipe was relatively high. The average HC conversion rate at the cold operation stage reached 92%, which was 2.3% higher than that of the GF pipe.

Figure 16 shows a comparison of the particulate matter emissions of the exhaust pipes between the different covering schemes at each stage during cold operation. Figure 16 shows that the particulate matter emissions of the BF pipe were the lowest.

## 4. Conclusions

This paper presented an experimental study on the thermal insulation performance of exhaust pipes coated with various materials (basalt and glass fiber materials) under different braiding forms (sleeve, winding and felt types) and the effects on the emission characteristics of diesel engines. The following conclusions can be drawn from this study.

The thermal insulation performance of the basalt fiber material is better than that of the glass fiber material. This occurs because the thermal conductivity of the basalt fiber material is lower than that of the glass fiber material. The average temperature decrease in the BF pipe throughout the entire cold WHTC cycle is the smallest, and its average temperature is 0.5 °C lower than that of the GF pipe, which is a small difference. During the 600 s period before the cold cycle, the thermal insulation performance of the basalt fiber material is obviously better than that of the glass fiber material, and the average temperature drop is 2.6 °C smaller than that of the GF pipe. The basalt fiber material of the felt covering type attains the best thermal insulation performance, and the average temperature drop is 2.6 °C and 2.9 °C smaller than that of the sleeve- and winding-type materials, respectively.

The gaseous pollutant emission performance of the after-treatment system coated with basalt fiber material is better than that of the after-treatment system coated with glass fiber material. Throughout the full WHTC cycle, during the transition phase (600~1200 s) and the thermal operation phase (1200~1800 s), the average conversion rates of NO_x_, CO and HCs of each pipe are obviously improved. The average conversion rates of NO_x_, CO, and HCs of the BF pipe are all the highest. Among them, the average NO_x_ conversion rate is 13.6%, 11% and 3% higher than that of the BS, BW and GF pipes, respectively. The BF pipe average CO conversion rate is 1.1%, 2.7%, and 4.3% higher, respectively, and the BF pipe average HC conversion rate is 6.9%, 8.3%, and 1.3% higher, respectively.

During the first 600 s of the cold operation stage, the gaseous pollutant emission performance of the basalt fiber-coated after-treatment system is notably higher than that of the glass fiber-coated after-treatment system. The NOx conversion rate of the BF pipe is 7.4% higher than that of the GF pipe. The CO conversion rate of the BF pipe during the first 100 s is 24.5% higher than that of the GF pipe, and the HC conversion rate of the BF pipe is 2.3% higher than that of the GF pipe. Little difference was observed in diesel particulate matter emissions between the exhaust pipes with the different covering schemes.

## Figures and Tables

**Figure 1 micromachines-12-01155-f001:**
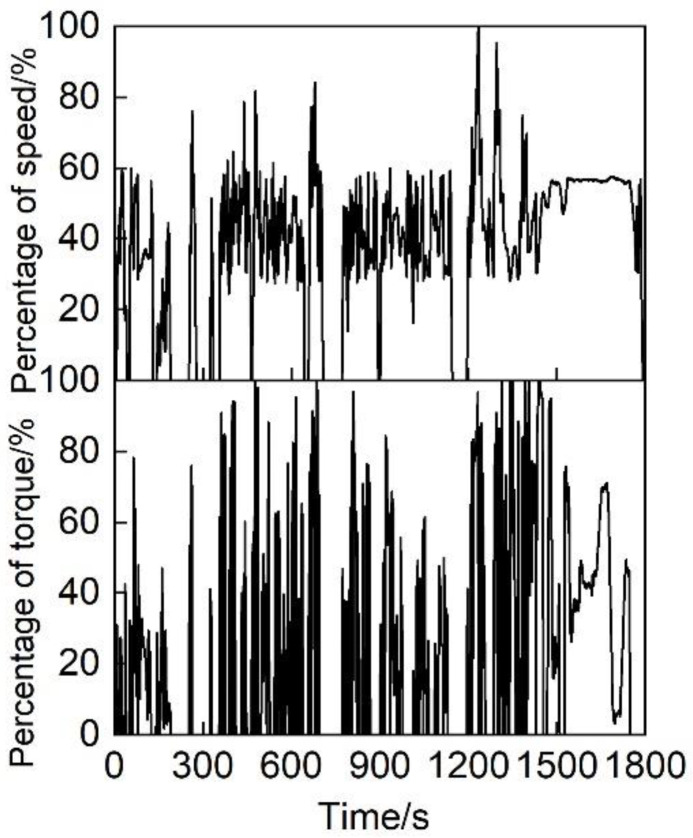
Schematic diagram of the WHTC cycle.

**Figure 2 micromachines-12-01155-f002:**
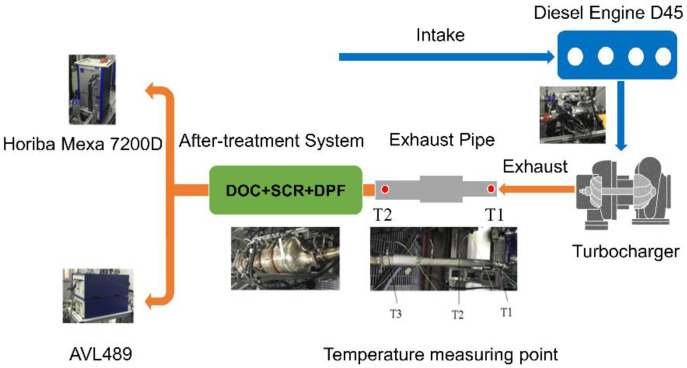
Layout of the test bench and temperature measurement points.

**Figure 3 micromachines-12-01155-f003:**
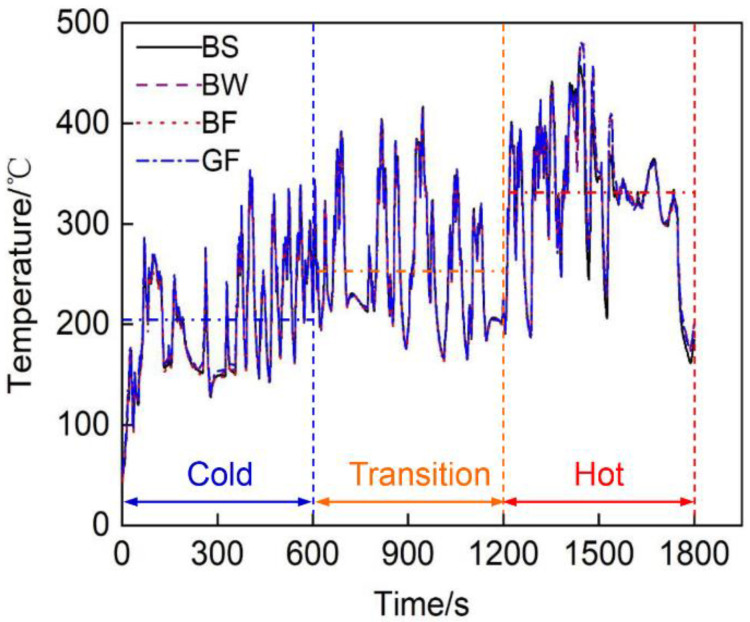
Temperature at the T1 measurement point of the exhaust pipes with different covering schemes over time.

**Figure 4 micromachines-12-01155-f004:**
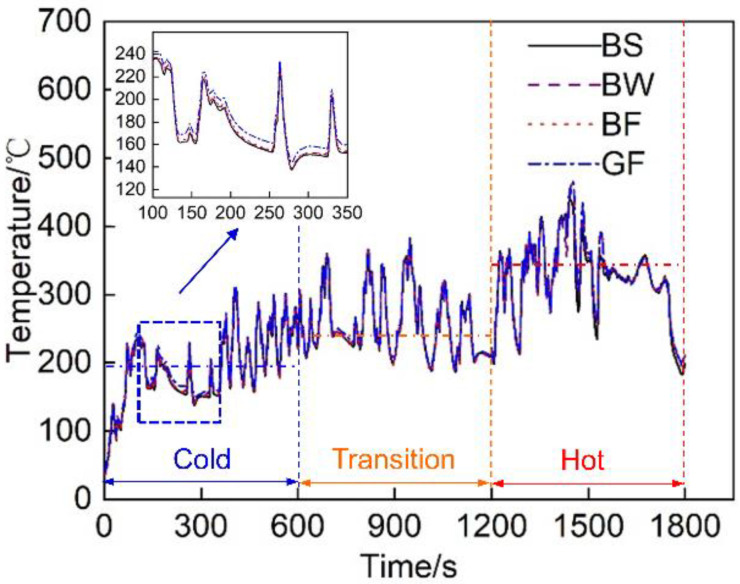
Temperature at the T2 measurement point of the exhaust pipes with different covering schemes over time.

**Figure 5 micromachines-12-01155-f005:**
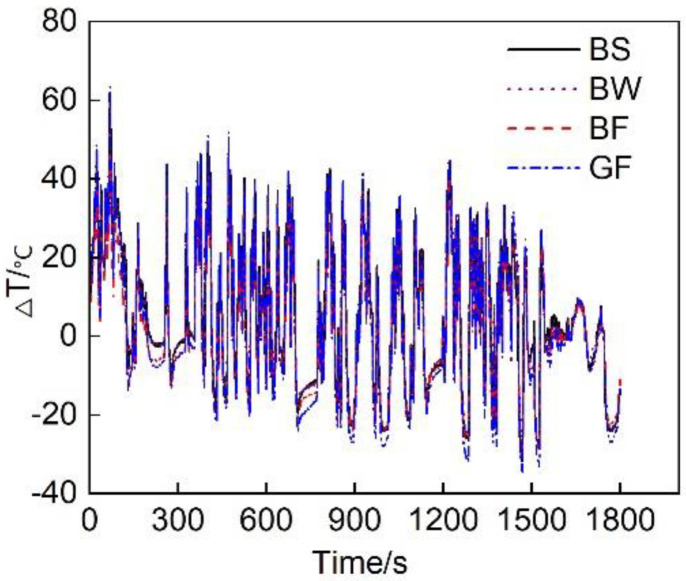
Temperature decrease Δ T in the exhaust pipes with different covering schemes over time.

**Figure 6 micromachines-12-01155-f006:**
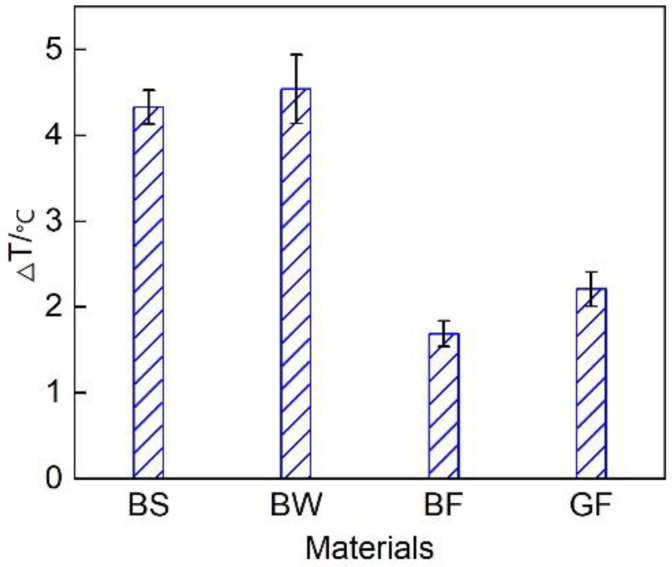
Average temperature decrease in the exhaust pipes with different covering schemes.

**Figure 7 micromachines-12-01155-f007:**
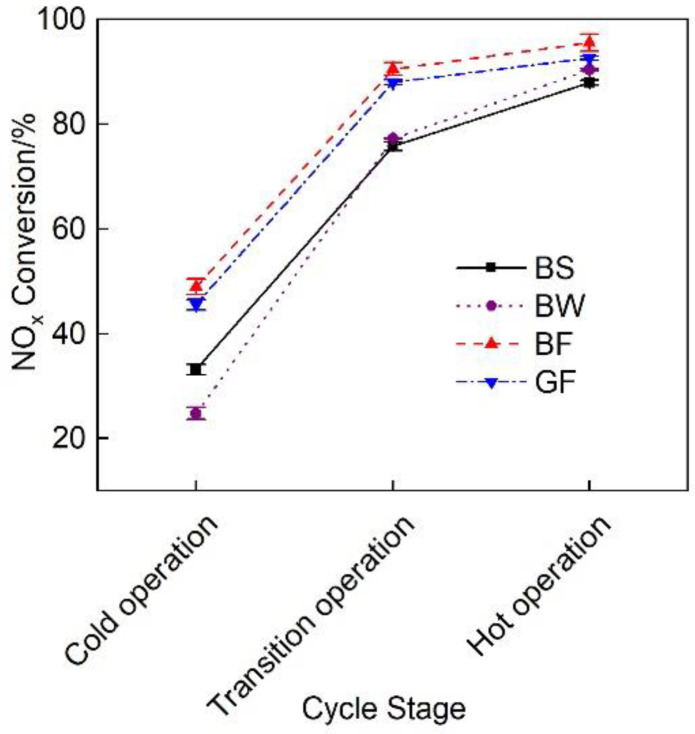
NO_x_ conversion rate of the different exhaust pipe covering schemes.

**Figure 8 micromachines-12-01155-f008:**
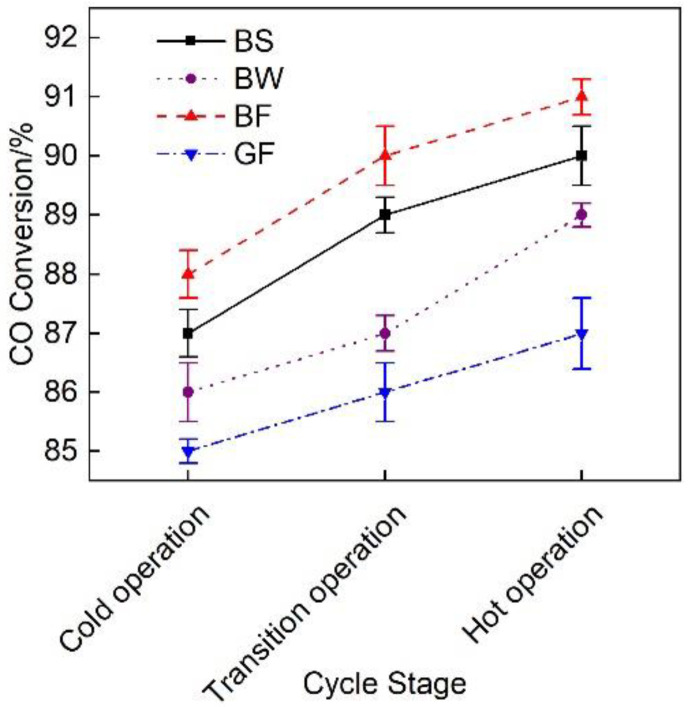
CO conversion rate of the different exhaust pipe covering schemes.

**Figure 9 micromachines-12-01155-f009:**
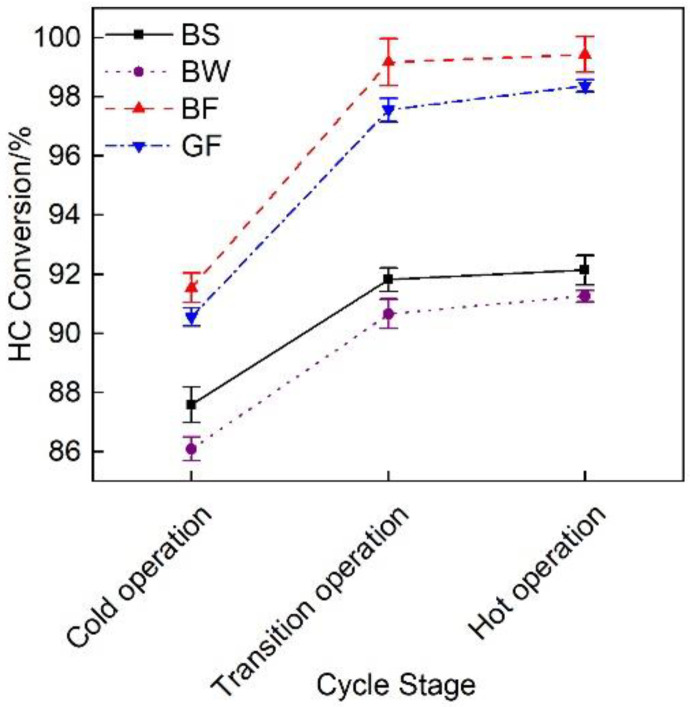
HC conversion rate of the different exhaust pipe covering schemes.

**Figure 10 micromachines-12-01155-f010:**
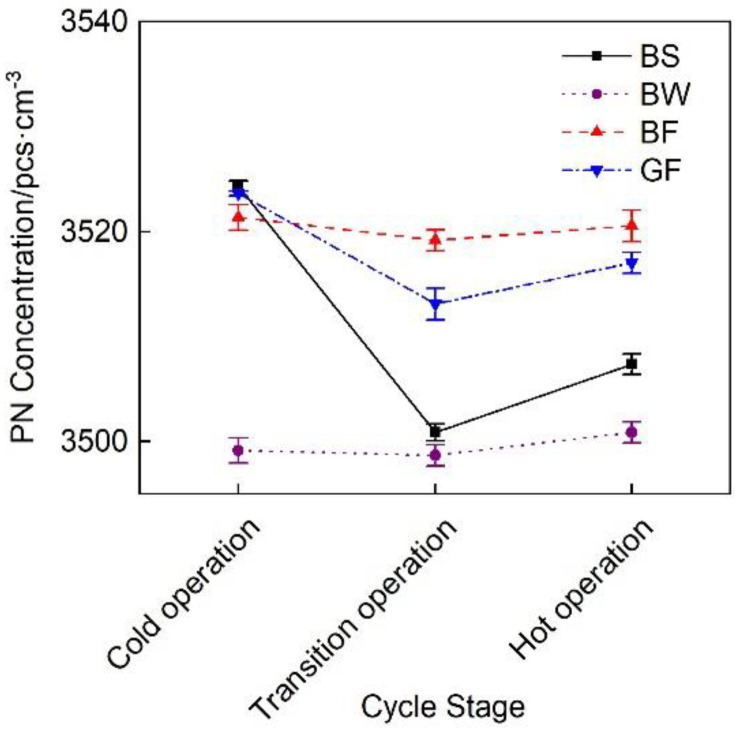
PN concentration change considering the different exhaust pipe covering schemes.

**Figure 11 micromachines-12-01155-f011:**
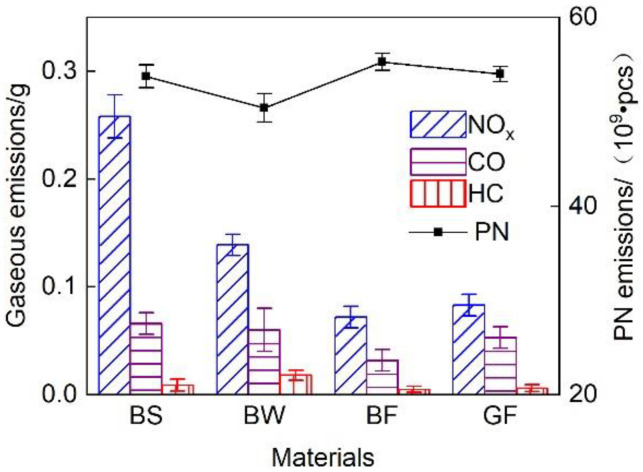
Total emissions during the WHTC cycle.

**Figure 12 micromachines-12-01155-f012:**
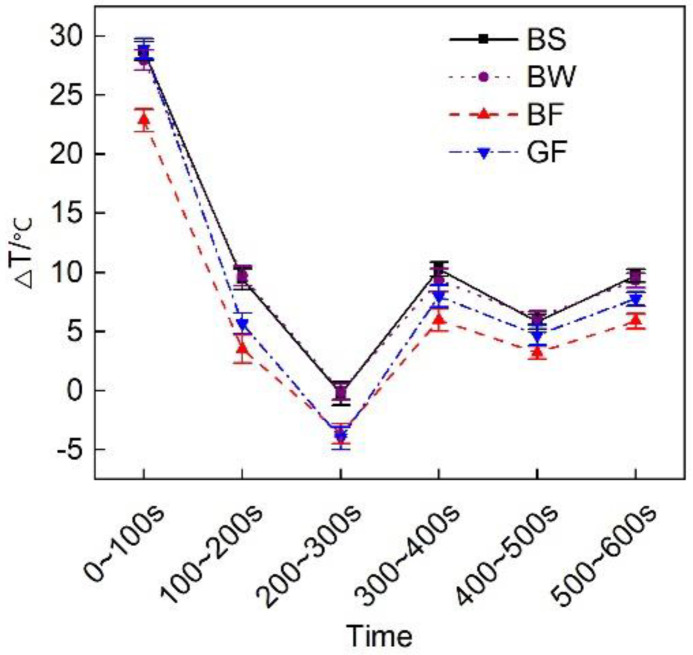
Average temperature drop at each substage of the initial idling stage.

**Figure 13 micromachines-12-01155-f013:**
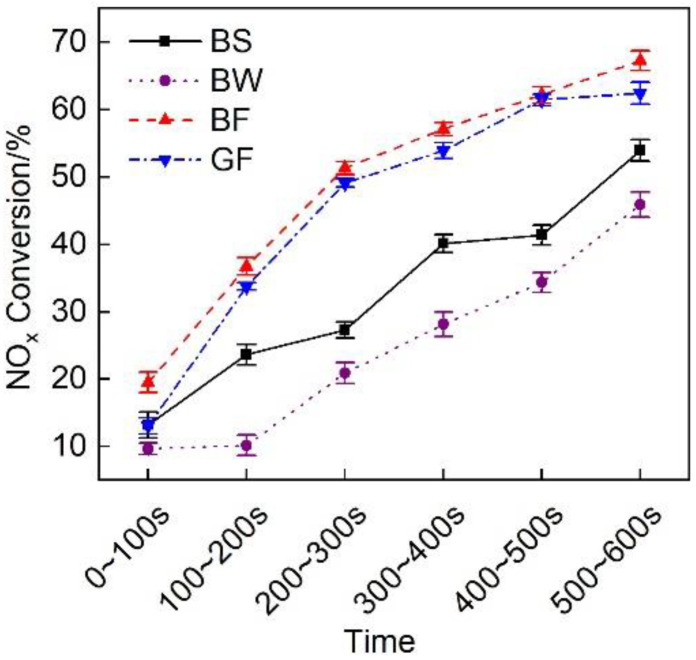
NO_x_ conversion efficiency at each substage of the initial idling stage of the different exhaust pipe covering schemes.

**Figure 14 micromachines-12-01155-f014:**
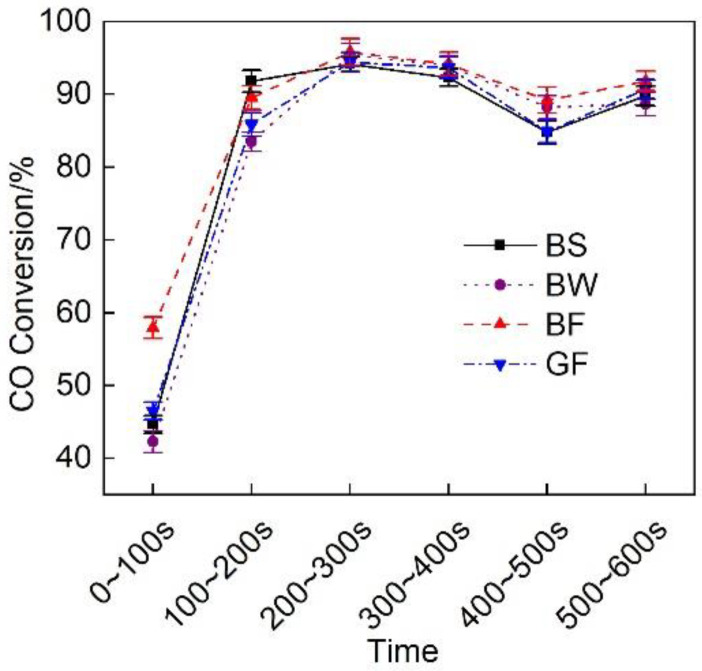
CO conversion efficiency at each substage of the initial idling stage of the different exhaust pipe covering schemes.

**Figure 15 micromachines-12-01155-f015:**
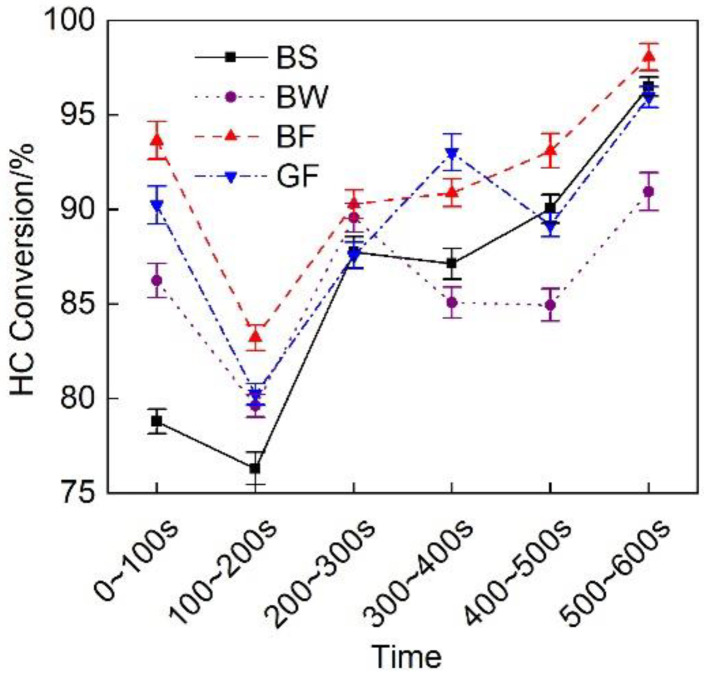
HC conversion efficiency of the different exhaust pipe covering schemes at each substage of the initial idling stage.

**Figure 16 micromachines-12-01155-f016:**
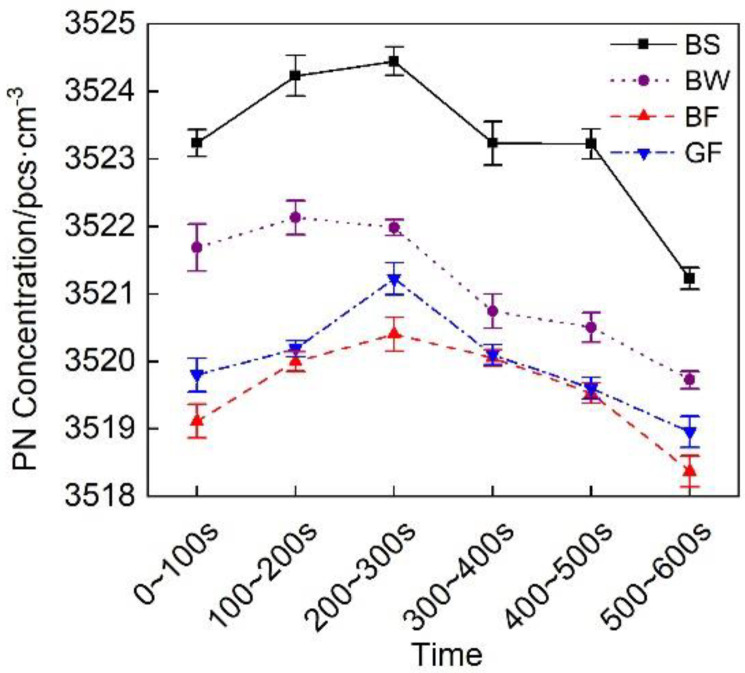
PN concentration of the different exhaust pipe covering schemes at each substage of the initial idling stage.

**Table 1 micromachines-12-01155-t001:** Test sample.

Serial Number	Prototype	Thickness (mm)	Bulk Density (kg·m^−3^)	Thermal Conductivity (W/m∙k)	End-Use Temperature (°C)	Morphology
1	BS	5	120	0.031	780	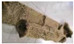
2	BW	5	120	0.031	780	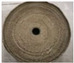
3	BF	5	120	0.031	780	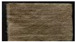
4	GF	5	120	0.049	400	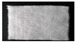

**Table 2 micromachines-12-01155-t002:** Parameters of the diesel engine.

Project	Parameter
Diesel engine model	D45
Displacement/L	4.5
Rated power/kW	150
Rated speed/r min^−1^	2300
After-treatment device	DOC + SCR + DPF

**Table 3 micromachines-12-01155-t003:** Sampling equipment and models.

Sample Item	Device Model	Sample Content
Gaseous substance	Horiba 7200D	NOx, CO, and hydrocarbons (HCs)
Number of particles (PN)	AVL489	PN

**Table 4 micromachines-12-01155-t004:** Test plan.

Exhaust Gas Temperature	Emission Data	Test Cycle
1. T1 before the pipe 2. T2 after the pipe	Gaseous substances: CO, HCs, and NOx Number of particles: PN	WHTC (cold state)

## Data Availability

The data used to support the findings of this study are included within the article.

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
