# Peer review of "Experimental Study on Diesel Engine Emission Characteristics Based on Different Exhaust Pipe Coating Schemes"

_micromachines, 2021, doi:10.3390/mi12101155_

Round 1

Reviewer 1 Report

Comments on the submitted article:

  1. Section 2.1. Test materials. More detailed information on the test exhaust pipe insulation materials should be provided.
  2. Is Device model: Horiba Mexa7X00 correctly described? The technical characteristics of the device should be provided.
  3. The methodology section should describe the methodology for calculating the conversion rate for different pollutants.
  4. 3, 4, 5 and other pictures must have better quality
  5. A more detailed analysis of the research results should be provided.

Reviewer 2 Report

Paper needs revision.

The materials used for covering pipes can't be referred as coating. The term coating should be replaced by covering.

The description of cover material should be provide in more detail. More clear images should be provided.

Data presented in fig. 6-16 has not been treated statistically. Error bars to be provided.

The figures need explanation especially in terms of fiber/fabric construction and related properties.
